# Multistep Distillation of Diffusion Models via Moment Matching

**Tim Salimans    Thomas Mensink    Jonathan Heek    Emiel Hoogeboom**

`{salimans,mensink,jheek,emielh}@google.com`

Google DeepMind, Amsterdam

## Abstract

We present a new method for making diffusion models faster to sample. The method distills many-step diffusion models into few-step models by matching conditional expectations of the clean data given noisy data along the sampling trajectory. Our approach extends recently proposed one-step methods to the multi-step case, and provides a new perspective by interpreting these approaches in terms of moment matching. By using up to 8 sampling steps, we obtain distilled models that outperform not only their one-step versions but also their original many-step teacher models, obtaining new state-of-the-art results on the Imagenet dataset. We also show promising results on a large text-to-image model where we achieve fast generation of high resolution images directly in image space, without needing autoencoders or upsamplers.

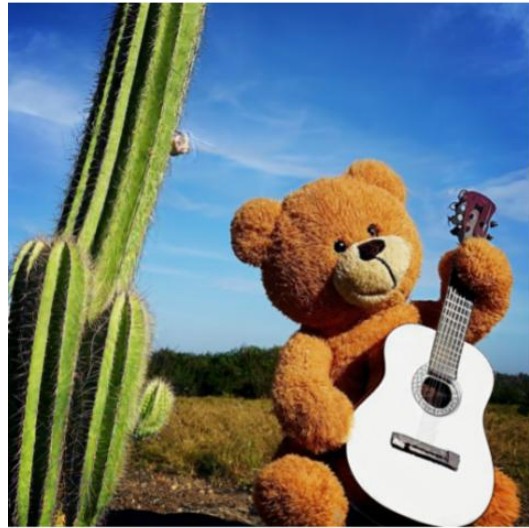
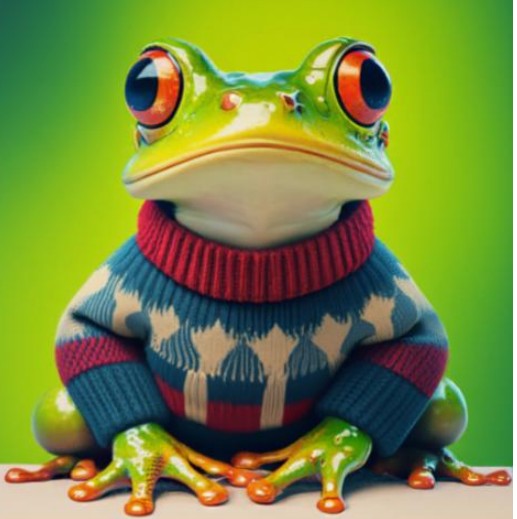

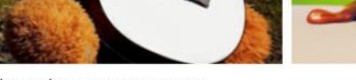

Figure 1: Selected 8-step samples from our distilled text-to-image model.

38th Conference on Neural Information Processing Systems (NeurIPS 2024).

# 1 Introduction

Diffusion models (Ho et al., 2020; Song & Ermon, 2019; Sohl-Dickstein et al., 2015) have recently become the state-of-the-art model class for generating images, video, audio, and other modalities. By casting the generation of high dimensional outputs as an iterative denoising process, these models have made the problem of learning to synthesize complex outputs tractable. Although this decomposition simplifies the training objective compared to alternatives like GANs, it shifts the computational burden to inference: Sampling from diffusion models usually requires hundreds of neural network evaluations, making these models expensive to use in applications.

To reduce the cost of inference, recent work has moved towards *distilling* diffusion models into generators that are faster to sample. The methods proposed so far can be subdivided into 2 classes: deterministic methods that aim to directly approximate the output of the iterative denoising process in fewer steps, and distributional methods that try to generate output with the same approximate distribution as learned by the diffusion model. Here we propose a new method for distilling diffusion models of the second type: We cast the problem of distribution matching in terms of matching conditional expectations of the clean data given the noisy data along the sampling trajectory of the diffusion process. The proposed method is closely related to previous approaches applying score matching with an auxiliary model to distilled one-step generators, but the moment matching perspective allows us to generalize these methods to the few-step setting where we obtain large improvements in output quality, even outperforming the many-step base models our distilled generators are learned from. Finally, the moment matching perspective allows us to also propose a second variant of our algorithm that eliminates the need for the auxiliary model in exchange for processing two independent minibatches per parameter update.

# 2 Background

## 2.1 Diffusion Models

Diffusion models are trained by learning to invert a noise process that gradually destroys data from a clean data sample $\boldsymbol{x}$ according to $\boldsymbol{z}_t = \alpha_t \boldsymbol{x} + \sigma_t \boldsymbol{\epsilon}_t$ with $\boldsymbol{z}_t \sim \mathcal{N}(0, \mathbf{I})$, where $\alpha_t, \sigma_t$ are monotonic functions of diffusion time $t \in [0, 1]$. The coefficients $\alpha_t, \sigma_t$ may be specified in multiple equivalent ways. Here, we use the variance preserving specification (Ho et al., 2020) that has $\sigma_t^2 = 1 - \alpha_t^2$, $\alpha_0 = \sigma_1 = 1$, and $\alpha_1 = \sigma_0 = 0$, such that we have that $\boldsymbol{z}_0 = \mathbf{x}$ and $\boldsymbol{z}_1 \sim \mathcal{N}(0, \mathbf{I})$. When using a different specification of the noise process we can always convert to the variance preserving specification by rescaling the data. The quantity of importance is thus the signal-to-noise ratio: $\text{SNR}(t) = \alpha_t^2/\sigma_t^2$, rather than the coefficients individually (Kingma et al., 2021). To invert the specified diffusion process, we can sample from the posterior distribution:

$$q(\boldsymbol{z}_s|\boldsymbol{z}_t, \boldsymbol{x}) = \mathcal{N}(\boldsymbol{z}_s|\mu_{t \to s}(\boldsymbol{z}_t, \boldsymbol{x}), \sigma_{t \to s}), \tag{1}$$

$$\text{with } \sigma_{t \to s}^2 = \left(\frac{1}{\sigma_s^2} + \frac{\alpha_{t|s}^2}{\sigma_{t|s}^2}\right)^{-1} \text{ and } \boldsymbol{\mu}_{t \to s} = \sigma_{t \to s}^2 \left(\frac{\alpha_{t|s}}{\sigma_{t|s}^2} \boldsymbol{z}_t + \frac{\alpha_s}{\sigma_s^2} \boldsymbol{x}\right). \tag{2}$$

To sample from a learned diffusion model, we replace $\boldsymbol{x}$ by a prediction from a neural network $\hat{\boldsymbol{x}} = g_\theta(\boldsymbol{z}_t, t)$ that is fit to the data by minimizing $\mathbb{E}_{t \sim p(t), \boldsymbol{z}_t, \boldsymbol{x} \sim q(\boldsymbol{z}_t, \boldsymbol{x})} w(t) \|\mathbf{x} - g_\theta(\mathbf{z}_t)\|^2$, with weighting function $w(t)$ and where $q(\boldsymbol{z}_t, \boldsymbol{x})$ denotes sampling $\boldsymbol{x}$ from the data and then producing $\boldsymbol{z}_t$ by forward diffusion. The sampling process starts with pure noise $\boldsymbol{z}_1 \sim \mathcal{N}(0, \mathbf{I})$ and iteratively denoises the data according to $q(\boldsymbol{z}_s|\boldsymbol{z}_t, \hat{\boldsymbol{x}})$ for a discrete number of timesteps $k$, following Algorithm 1.

If we attain the optimal solution $\hat{\boldsymbol{x}} = \mathbb{E}[\boldsymbol{x}|\boldsymbol{z}_t]$ and let $k \to \infty$ the sampling process becomes exact, then the learned diffusion model can be shown to be a universal distribution approximator (Song et al., 2021b). To get close to this ideal, $k$ typically needs to be quite large, making diffusion models a very computationally expensive class of models (Luccioni et al., 2023).

## 2.2 Generalized method of moments

An alternative to the well-known maximum likelihood estimation method is the method of moments, also known as moment matching. Traditionally for univariate distributions, one matches moments $m_k = \mathbb{E}_{x \sim p_X}[x^k]$ of a random variable $X$. The canonical example is a Gaussian distribution, which is defined by the first two moments (i.e. the mean and variance) and all (centered) higher order moments

**Algorithm 1** Ancestral sampling algorithm used for both standard denoising diffusion models as well as our distilled models. For standard models typically $256 \leq k \leq 1000$, for distilled $1 \leq k \leq 16$.

---

**Require:** Denoising model $g_\theta(\mathbf{z}_t, t)$, number of sampling steps $k$
    Initialize noisy data $\mathbf{z}_1 \sim N(0, \mathbf{I})$
    **for** $t \in \{1, (k-1)/k, \ldots, 2/k, 1/k\}$ **do**
        Predict clean data using $\hat{\mathbf{x}} = g_\theta(\mathbf{z}_t, t)$
        Set next timestep $s = t - 1/k$
        Sample next noisy data point $\mathbf{z}_s \sim q(\mathbf{z}_s | \mathbf{z}_t, \hat{\mathbf{x}})$
    **end for**
    Return approximate sample $\hat{\mathbf{x}}$

---

are zero. Fitting a distribution by setting its moments equal to the moments of the data is then a consistent parameter estimation method, and can be readily extended to multivariate distributions, e.g. by matching the mean and covariance matrix for a multivariate Gaussian.

One can generalize the method of moments to arbitrary high dimensional functions $f : \mathbb{R}^d \to \mathbb{R}^k$ and match the moment vector $\boldsymbol{m}$ as defined by: $\boldsymbol{m} = \mathbb{E}_{\boldsymbol{x} \sim p_X}[f(\boldsymbol{x})]$, which is called the *Generalized Method of Moments* (GMM, Hansen (1982)). Matching such moments can be done by minimizing a distance between the moments such as $||\mathbb{E}_{\boldsymbol{x} \sim p_{\boldsymbol{\theta}}} f(\boldsymbol{x}) - \mathbb{E}_{\boldsymbol{x} \sim p_X} f(\boldsymbol{x})||^2$ where $p_{\boldsymbol{\theta}}$ is the generative model and $p_X$ the data distribution. The distillation method we propose in the next section can be interpreted as a special case of this class of estimation methods.

## 3 Moment Matching Distillation

Many-step sampling from diffusion models starts by initializing noisy data $\mathbf{z}_1 \sim N(0, \mathbf{I})$, which is then iteratively refined by predicting the clean data using $\hat{\mathbf{x}} = g_\theta(\mathbf{z}_t, t)$, and sampling a slightly less noisy data point $\mathbf{z}_s \sim q(\mathbf{z}_s | \mathbf{z}_t, \hat{\mathbf{x}})$ for new timestep $s < t$, until the final sample is obtained at $s = 0$, as described is described in Algorithm 1. If $\hat{\mathbf{x}} = \mathbb{E}_q[\mathbf{x} | \mathbf{z}_t]$ this procedure is guaranteed to sample from the data distribution $q(\mathbf{x})$ if the number of sampling steps grows infinitely large. Here we aim to achieve a similar result while taking many fewer sampling steps than would normally be required. To achieve this we finetune our denoising model $g_\theta$ into a new model $g_\eta(\mathbf{z}_t, t)$ which we sample from using the same algorithm, but with a strongly reduced number of sampling steps $k$, for say $1 \leq k \leq 8$.

To make our model produce accurate samples for a small number of sampling steps $k$, the goal is now no longer for $\tilde{\mathbf{x}} = g_\eta(\mathbf{z}_t, t)$ to approximate the expectation $\mathbb{E}_q[\mathbf{x} | \mathbf{z}_t]$ but rather to produce an approximate sample from this distribution. In particular, if $\tilde{\mathbf{x}} \sim q(\mathbf{x} | \mathbf{z}_t)$ then Algorithm 1 produces exact samples from the data distribution $q$ for any choice of the number of sampling steps. If $g_\eta$ perfectly approximates $q(\mathbf{x} | \mathbf{z}_t)$ as intended, we have that

$$\mathbb{E}_{\mathbf{x} \sim q(\mathbf{x}), \mathbf{z}_t \sim q(\mathbf{z}_t | \mathbf{x}), \tilde{\mathbf{x}} \sim g_\eta(\mathbf{z}_t), \mathbf{z}_s \sim q(\mathbf{z}_s | \mathbf{z}_t, \tilde{\mathbf{x}})}[\tilde{\mathbf{x}} | \mathbf{z}_s] = \mathbb{E}_{\mathbf{x} \sim q(\mathbf{x}), \mathbf{z}_s \sim q(\mathbf{z}_s | \boldsymbol{x})}[\mathbf{x} | \mathbf{z}_s]$$
$$\mathbb{E}_g[\tilde{\mathbf{x}} | \mathbf{z}_s] = \mathbb{E}_q[\mathbf{x} | \mathbf{z}_s]. \tag{3}$$

In words: The conditional expectation of clean data should be identical between the data distribution $q$ and the sampling distribution $g$ of the distilled model.

Equation 3 gives us a set of moment conditions that uniquely identifies the target distribution, similar to how the regular diffusion training loss identifies the data distribution (Song et al., 2021b). These moment conditions can be used as the basis of a distillation method to finetune $g_\eta(\mathbf{z}_t, t)$ from the denoising model $g_\theta$. In particular, we can fit $g_\eta$ to $q$ by minimizing the L2-distance between these moments:

$$\tilde{L}(\eta) = \frac{1}{2} \mathbb{E}_{g(\mathbf{z}_s)} ||\mathbb{E}_g[\tilde{\mathbf{x}} | \mathbf{z}_s] - \mathbb{E}_q[\mathbf{x} | \mathbf{z}_s]||^2. \tag{4}$$

In practice, we evaluate the moments using a sample $\boldsymbol{z}_s$ from our generator distribution, but do not incorporate its dependence on the parameters $\eta$ when calculating gradients of the loss. This decision is purely empirical, as we find it results in more stable training compared to using the full gradient. The approximate gradient of $\tilde{L}(\eta)$ is then given by

$$\left(\nabla_\eta \mathbb{E}_g[\tilde{\mathbf{x}} | \mathbf{z}_s]\right)^T (\mathbb{E}_g[\tilde{\mathbf{x}} | \mathbf{z}_s] - \mathbb{E}_q[\mathbf{x} | \mathbf{z}_s]) + \nabla_\eta \left(\tfrac{1}{2} \mathbb{E}_q[\mathbf{x} | \mathbf{z}_s]^T \mathbb{E}_q[\mathbf{x} | \mathbf{z}_s]\right) \approx \left(\nabla_\eta \tilde{\mathbf{x}}\right)^T (\mathbb{E}_g[\tilde{\mathbf{x}} | \mathbf{z}_s] - \mathbb{E}_q[\mathbf{x} | \mathbf{z}_s]), \tag{5}$$

where we approximate the first expectation using a single Monte-Carlo sample $\tilde{\mathbf{x}}$ and where the second term is zero as it does not depend on $g_\eta$. Following this approximate gradient is then equivalent to minimizing the loss

$$L(\eta) = \mathbb{E}_{\mathbf{z}_t \sim q(\mathbf{z}_t), \tilde{\mathbf{x}} \sim g_\eta(\mathbf{z}_t), \mathbf{z}_s \sim q(\mathbf{z}_s | \mathbf{z}_t, \tilde{\mathbf{x}})}[\tilde{\mathbf{x}}^T \mathrm{sg}(\mathbb{E}_g[\tilde{\mathbf{x}} | \mathbf{z}_s] - \mathbb{E}_q[\mathbf{x} | \mathbf{z}_s])], \tag{6}$$

where sg denotes stop-gradient. This loss is minimized if $\mathbb{E}_g[\tilde{\mathbf{x}} | \mathbf{z}_s] = \mathbb{E}_q[\mathbf{x} | \mathbf{z}_s]$ as required. Unfortunately, the expectation $\mathbb{E}_g[\tilde{\mathbf{x}} | \mathbf{z}_s]$ is not analytically available, which makes the direct application of Equation 6 impossible. We therefore explore two variations on this moment matching procedure: In Section 3.1 we approximate $\mathbb{E}_g[\tilde{\mathbf{x}} | \mathbf{z}_s]$ by a second denoising model, and in Section 3.2 we instead apply moment matching directly in parameter space rather than $\mathbf{x}$-space.

## 3.1 Alternating optimization of the moment matching objective

Our first approach to calculating the moment matching objective in equation 6 is to approximate $\mathbb{E}_g[\tilde{\mathbf{x}} | \mathbf{z}_s]$ with an auxiliary denoising model $g_\phi$ trained using a standard diffusion loss on samples from our generator model $g_\eta$. We then update $g_\phi$ and $g_\eta$ in alternating steps, resulting in Algorithm 2.

---

**Algorithm 2** Moment matching algorithm with *alternating* optimization of generator $g_\eta$ and auxiliary denoising model $g_\phi$.

---

**Require:** Pretrained denoising model $g_\theta(\mathbf{z}_t)$, generator $g_\eta$ to distill, auxiliary denoising model $g_\phi$, number of sampling steps $k$, time sampling distribution $p(s)$, loss weight $w(s)$, and dataset $\mathcal{D}$.
  **for** $n = 0$:N **do**
    Sample target time $s \sim p(s)$, sample time delta $\delta_t \sim U[0, 1/k]$.
    Set sampling time $t = \mathrm{minimum}(s + \delta_t, 1)$.
    Sample clean data from $\mathcal{D}$ and do forward diffusion to produce $\mathbf{z}_t$.
    Sample $\mathbf{z}_s$ from the distilled generator using $\tilde{\mathbf{x}} = g_\eta(\mathbf{z}_t), \mathbf{z}_s \sim q(\mathbf{z}_s | \mathbf{z}_t, \tilde{\mathbf{x}})$.
    **if** $n$ is even **then**
      Minimize $L(\phi) = w(s)\{\|\tilde{\mathbf{x}} - g_\phi(\mathbf{z}_s)\|^2 + \|g_\theta(\mathbf{z}_s) - g_\phi(\mathbf{z}_s)\|^2\}$ w.r.t. $\phi$
    **else**
      Minimize $L(\eta) = w(s)\tilde{\mathbf{x}}^T \mathrm{sg}[g_\phi(\mathbf{z}_s) - g_\theta(\mathbf{z}_s)]$ w.r.t. $\eta$
    **end if**
  **end for**

---

Here we have chosen to train our generator $g_\eta$ on all continuous times $t \in (0, 1]$ even though at inference time (Algorithm 1) we only evaluate on $k$ discrete timesteps. Similarly we train with randomly sampled time delta $\delta_t$ rather than fixing this to a single value. These choices were found to increase the stability and performance of the proposed algorithm. Further, we optimize $g_\phi$ not just to predict the sampled data $\tilde{\mathbf{x}}$ but also regularize it to stay close to the teacher model $g_\theta$: On convergence this would cause $g_\phi$ to predict the average of $\tilde{\mathbf{x}}$ and $g_\theta$, which has the effect of multiplying the generator loss $L(\eta)$ by $1/2$ compared to the loss we introduced in Equation 6.

The resulting algorithm resembles the alternating optimization of a GAN (Goodfellow et al., 2020), and like a GAN is generally not guaranteed to converge. In practice, we find that Algorithm 2 is stable for the right choice of hyperparameters, especially when taking $k \geq 8$ sampling steps. The algorithm also closely resembles *Variational Score Distillation* as previously used for distilling 1-step generators $g_\eta$ in *Diff-Instruct*. We discuss this relationship in Section 4.

## 3.2 Parameter-space moment matching

Alternating optimization of the moment matching objective (Algorithm 2) is difficult to analyze theoretically, and the requirement to keep track of two different models adds engineering complexity. We therefore also experiment with an *instantaneous* version of the auxiliary denoising model $g_{\phi^*}$, where $\phi^*$ is determined using a single infinitesimal gradient descent step on $L(\phi)$ (defined in Algorithm 2), evaluated on a single minibatch. Starting from teacher parameters $\theta$, and preconditioning the loss gradient with a pre-determined scaling matrix $\Lambda$, we can define:

$$\phi(\lambda) \equiv \theta - \lambda\Lambda\nabla_\phi L(\phi)|_{\phi=\theta}, \text{ so that } \phi^* = \lim_{\lambda \to 0} \phi(\lambda). \tag{7}$$

Now we use $\phi(\lambda)$ in calculating $L(\eta)$ from Algorithm 2, take the first-order Taylor expansion for $g_{\phi(\lambda)}(\mathbf{z}_s) - g_\theta(\mathbf{z}_s) \approx \lambda \frac{\partial g_\theta(\mathbf{z}_s)}{\partial \theta}(\phi(\lambda) - \theta) = \lambda \frac{\partial g_\theta(\mathbf{z}_s)}{\partial \theta} \Lambda \nabla_\phi L(\phi)|_{\phi=\theta}$, and scale the loss with the inverse of $\lambda$ to get:

$$L_{\text{instant}}(\eta) = \lim_{\lambda \to 0} \frac{1}{\lambda} L_{\phi(\lambda)}(\eta) = w(s)\tilde{\mathbf{x}}^T \text{sg}\Big\{ \frac{\partial g_\theta(\mathbf{z}_s)}{\partial \theta} \Lambda \nabla_\phi L(\phi)|_{\phi=\theta}\Big\}, \tag{8}$$

where $\frac{\partial g_\theta(\mathbf{z}_s)}{\partial \theta}$ is the Jacobian of $g_\theta$, and where $\nabla_\phi L(\phi)$ is evaluated on an independent minibatch from $\tilde{\mathbf{x}}$ and $\mathbf{z}_s$. In modern frameworks for automatic differentiation, like JAX (Bradbury et al., 2018), the quantity within the curly braces can be most easily expressed using specialized functions for calculating Jacobian-vector products.

The loss can now equivalently be expressed as performing moment matching in teacher-parameter space rather than $\mathbf{x}$-space. Denoting $L_\theta(\mathbf{x}, \mathbf{z}_s) \equiv w(s)\|\mathbf{x} - g_\theta(\mathbf{z}_s)\|^2$, and letting $\tilde{L}_{\text{instant}}(\eta) = L_{\text{instant}}(\eta) + \text{constant}$, we have (as derived fully in Appendix A):

$$\tilde{L}_{\text{instant}}(\eta) \equiv \frac{1}{2}\|\mathbb{E}_{\mathbf{z}_t \sim q, \tilde{\mathbf{x}}=g_\eta(\mathbf{z}_t), \mathbf{z}_s \sim q(\mathbf{z}_s|\mathbf{z}_t, \tilde{\mathbf{x}})} \nabla_\theta L_\theta(\tilde{\mathbf{x}}, \text{sg}(\mathbf{z}_s))\|_\Lambda^2 \tag{9}$$

$$= \frac{1}{2}\|\mathbb{E}_{\mathbf{z}_t \sim q, \tilde{\mathbf{x}}=g_\eta(\mathbf{z}_t), \mathbf{z}_s \sim q(\mathbf{z}_s|\mathbf{z}_t, \tilde{\mathbf{x}})} \nabla_\theta L_\theta(\tilde{\mathbf{x}}, \text{sg}(\mathbf{z}_s)) - \mathbb{E}_{\mathbf{x}, \mathbf{z}_s' \sim q} \nabla_\theta L_\theta(\mathbf{x}, \mathbf{z}_s')\|_\Lambda^2, \tag{10}$$

where the gradient of the teacher training loss is zero when sampling from the training distribution, $\mathbb{E}_{\mathbf{x}, \mathbf{z}_s' \sim q} \nabla_\theta L_\theta(\mathbf{x}, \mathbf{z}_s') = 0$, if the teacher attained a minimum of its training loss.

The instantaneous version of our moment matching loss can thus be interpreted as trying to match teacher gradients between the training data and generated data. This makes it a special case of the *Efficient Method of Moments* (Gallant & Tauchen, 1996), a classic method in statistics where a teacher model $p_\theta$ is first estimated using maximum likelihood, after which its gradient is used to define a moment matching loss for learning a second model $g_\eta$. Under certain conditions, the second model then attains the statistical efficiency of the maximum likelihood teacher model. The difference between our version of this method and that proposed by Gallant & Tauchen (1996) is that in our case the loss of the teacher model is a weighted denoising loss, rather than the log-likelihood of the data.

The moment matching loss $\tilde{L}_{\text{instant}}(\eta)$ is minimized if the teacher model has zero loss gradient when evaluated on data generated by the distilled student model $g_\eta$. In other words, optimization is successful if the teacher model cannot see the difference between real and generated data and would not change its parameters when trained on the generated data. We summarize the practical implementation of moment matching in parameter-space in Algorithm 3 and Figure 2.

---

**Algorithm 3** Parameter-space moment matching algorithm with *instant* denoising model $g_{\phi^*}$.

---

**Require:** Pretrained denoising model $g_\theta(\mathbf{z}_t)$, generator $g_\eta$ to distill, gradient scaling matrix $\Lambda$, number of sampling steps $k$, time sampling distribution $p(s)$, loss weight $w(s)$, and dataset $\mathcal{D}$.
  **for** $n = 0$:N **do**
    Sample target time $s \sim p(s)$, sample time delta $\delta_t \sim U[0, 1/k]$.
    Set sampling time $t = \text{minimum}(s + \delta_t, 1)$.
    Sample two independent batches of data from $\mathcal{D}$ and do forward diffusion to produce $\mathbf{z}_t, \mathbf{z}_t'$.
    For both batches sample $\mathbf{z}_s, \mathbf{z}_s'$ from the distilled generator using $\tilde{\mathbf{x}} = g_\eta(\mathbf{z}_t), \mathbf{z}_s \sim q(\mathbf{z}_s|\mathbf{z}_t, \tilde{\mathbf{x}})$.
    Evaluate teacher gradient on one batch: $\nu = \Lambda \nabla_\theta L_\theta(\tilde{\mathbf{x}}', \mathbf{z}_s')$
    On the other batch, minimize $L_{\text{instant}}(\eta) = w(s)\tilde{\mathbf{x}}^T \text{sg}\Big\{ \frac{\partial g_\theta(\mathbf{z}_s)}{\partial \theta} \nu \Big\}$ w.r.t. $\eta$
  **end for**

---

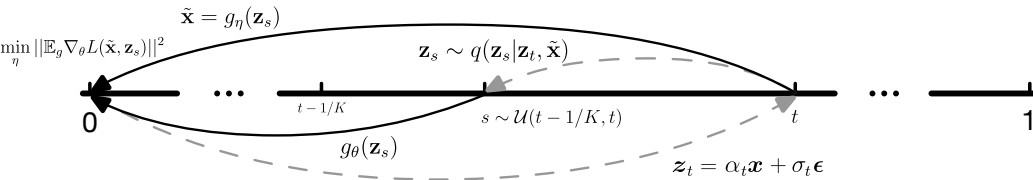

Figure 2: Visualization of Algorithm 3: Moment matching in parameter space starts with applying forward diffusion to data from our dataset, mapping this to clean samples using the distilled generator model, and then minimizes the gradient of the teacher loss on this generated data.

### 3.3 Hyperparameter choices

In our choice of hyperparameters we choose to stick as closely as possible to the values recommended in EDM (Karras et al., 2022), some of which were also used in Diff-Instruct (Luo et al., 2024) and DMD (Yin et al., 2023). We use the EDM test time noise schedule for $p(s)$, as well as their training loss weighting for $w(s)$, but we shift all log-signal-to-noise ratios with the resolution of the data following Hoogeboom et al. (2023). For our gradient preconditioner $\Lambda$, as used in Section 3.2, we use the preconditioner defined in Adam (Kingma & Ba, 2014), which can be loaded from the teacher checkpoint or calculated fresh by running a few training steps before starting distillation. During distillation, $\Lambda$ is not updated.

To get stable results for small numbers of sampling steps ($k = 1, 2$) we find that we need to use a weighting function $w(s)$ with less emphasis on high-signal (low $s$) data than in the EDM weighting. Using a flat weight $w(s) = 1$ or the adaptive weight from DMD (Yin et al., 2023) works well.

As with previous methods, it's possible to enable classifier-free guidance (Ho & Salimans, 2022) when evaluating the teacher model $g_\theta$. We find that guidance is typically not necessary if output quality is measured by FID, though it does increase Inception Score and CLIP score. To enable classifier-free guidance and prediction clipping for the teacher model in Algorithm 3, we need to define how to take gradients through these modifications: Here we find that a simple straight-through approximation works well, using the backward pass of the unmodified teacher model.

## 4 Related Work

In the case of one-step sampling, our method in Algorithm 2 is a special case of *Variational Score Distillation*, *Diff-Instruct*, and related methods (Wang et al., 2024; Luo et al., 2024; Yin et al., 2023; Nguyen & Tran, 2023) which distill a diffusion model by approximately minimizing the KL divergence between the distilled generator and the teacher model:

$$
\begin{aligned}
L(\eta) &= \mathbb{E}_{p(s)}[w(s)D_{\mathrm{KL}}(p_\eta(\mathbf{z}_s)|p_\theta(\mathbf{z}_s))] &\text{(11)}\\
&\approx \mathbb{E}_{p(s),p_\eta(\mathbf{z}_s)}[(w(s)/2\sigma_s^2)(\|\mathbf{z}_s - \alpha\mathrm{sg}[\hat{\mathbf{x}}_\theta(\mathbf{z}_s)]\|^2 - \|\mathbf{z}_s - \alpha\mathrm{sg}[\hat{\mathbf{x}}_\phi(\mathbf{z}_s)]\|^2)] &\text{(12)}\\
&= \mathbb{E}_{p(t),p_\eta(\mathbf{z}_s)}[(w(s)\alpha_s^2/\sigma_s^2)\tilde{\mathbf{x}}^T\mathrm{sg}[\hat{\mathbf{x}}_\phi(\mathbf{z}_s) - \hat{\mathbf{x}}_\theta(\mathbf{z}_s)]] + \text{constant} &\text{(13)}
\end{aligned}
$$

Here sg again denotes stop gradient, $p_\eta(\mathbf{z}_s)$ is defined by sampling $\tilde{\mathbf{x}} = g_\eta(\mathbf{z}_1)$ with $\mathbf{z}_1 \sim N(0, \mathbf{I})$, and $\mathbf{z}_s \sim q(\mathbf{z}_s|\tilde{\mathbf{x}})$ is sampled using forward diffusion starting from $\tilde{\mathbf{x}}$. The auxiliary denoising model $\hat{\mathbf{x}}_\phi$ is fit by minimizing $\mathbb{E}_{g_\eta}\tilde{w}(\mathbf{z}_s)\|\tilde{\mathbf{x}} - \hat{\mathbf{x}}_\phi(\mathbf{z}_s)\|^2$, which can be interpreted as score matching because $\mathbf{z}_s$ is sampled using forward diffusion started from $\tilde{\mathbf{x}}$. In our proposed algorithm, we sample $\mathbf{z}_s$ from the conditional distribution $q(\mathbf{z}_s|\tilde{\mathbf{x}}, \mathbf{z}_t)$: If $\mathbf{z}_t = \mathbf{z}_1 \sim N(0, \mathbf{I})$ is assumed to be fully independent of $\tilde{\mathbf{x}}$, i.e. that $\alpha_1^2/\sigma_1^2 = 0$, we have that $q(\mathbf{z}_s|\tilde{\mathbf{x}}, \mathbf{z}_1) = q(\mathbf{z}_s|\tilde{\mathbf{x}})$ so the two methods are indeed the same. However, this correspondence does not extend to the multi-step case: When we sample $\mathbf{z}_s$ from $q(\mathbf{z}_s|\mathbf{z}_t, \tilde{\mathbf{x}})$ for $\alpha_t^2/\sigma_t^2 > 0$, fitting $\hat{\mathbf{x}}_\phi$ through minimizing $\mathbb{E}_{g_\eta}\tilde{w}(\mathbf{z}_s)\|\tilde{\mathbf{x}} - \hat{\mathbf{x}}_\phi(\mathbf{z}_s)\|^2$ no longer corresponds to score matching. One could imagine fitting $\hat{\mathbf{x}}_\phi$ through score matching against the conditional distribution $q(\mathbf{z}_s|\mathbf{z}_t, \tilde{\mathbf{x}})$ but this did not work well when we tried it (see Appendix D for more detail). Instead, our moment matching perspective offers a justification for extending this class of distillation methods to the multistep case without changing the way we fit $\hat{\mathbf{x}}_\phi$. Indeed, we find that moment matching distillation also works when using deterministic samplers like DDIM (Song et al., 2021a) which also do not fit with the score matching perspective.

In addition to the one-step distillation methods based on score matching, our method is also closely related to adversarial multistep distillation methods, such as Xiao et al. (2021) and Xu et al. (2023a) which use the same conditional $q(\mathbf{z}_s|\mathbf{z}_t, \tilde{\mathbf{x}})$ we use. These methods train a discriminator model to tell apart data generated from the distilled model ($g_\eta$) from data generated from the base model ($g_\theta$). This discriminator is then used to define an adversarial divergence which is minimized w.r.t. $g_\eta$:

$$
L(\eta) = \mathbb{E}_{t\sim p(t), \mathbf{z}_t \sim q(\mathbf{z}_t, t)} D_{\mathrm{adv}}(p_\eta(\mathbf{z}_t)|p_\theta(\mathbf{z}_t)). \tag{14}
$$

The methods differ in their exact formulation of the adversarial divergence $D_{\mathrm{adv}}$, in the sampling of time steps, and in the use of additional losses. For example Xu et al. (2023a) train unconditional discriminators $D_\phi(\cdot, t)$ and decompose the adversarial objective in a marginal (used in the discriminator) and a conditional distribution approximated with an additional regression model. Xiao et al. (2021) instead use a conditional discriminator of the form $D_\phi(\cdot, \mathbf{z}_t, t)$.

# 5   Experiments

We evaluate our proposed methods in the class-conditional generation setting on the ImageNet dataset (Deng et al., 2009), which is the most well-established benchmark for comparing image quality. On this dataset we also run several ablations to show the effect of classifier-free guidance and other hyperparameter choices on our method. Finally, we present an experiment with a large text-to-image model to show our approach can also be scaled to this setting.

## 5.1   Class-conditional generation on ImageNet

We begin by evaluating on class-conditional ImageNet generation, at the $64 \times 64$ and $128 \times 128$ resolutions (Tables 1 and 2). Our results here are for a relatively small model with 400 million parameters based on *Simple Diffusion* (Hoogeboom et al., 2023). We distill our models for a maximum of 200,000 steps at batch size 2048, calculating FID every 5,000 steps. We report the optimal FID seen during the distillation process, keeping evaluation data and random seeds fixed across evaluations to minimize bias.

For our base models we report results with slight classifier-free guidance of $w = 0.1$, which gives the optimal FID. We also use an optimized amount of sampling noise, following Salimans & Ho (2022), which is slightly higher compared to equation 2. For our distilled models we obtained better results without classifier-free guidance, and we use standard ancestral sampling without tuning the sampling noise. We use identical hyperparameters across all our experiments. We compare against various distillation methods from the literature, including both distillation methods that produce deterministic samplers (progressive distillation, consistency distillation) and stochastic samplers (Diff-Instruct, adversarial methods).

Ranking the different methods by FID, we find that our moment matching distillation method is especially competitive when using $8+$ sampling steps, where it sets new state-of-the-art results, beating out even the best undistilled models using more than 1000 sampling steps, as well as its teacher model. For 1 sampling step some of the other methods show better results: Further improvement is likely possible by separately optimizing hyperparameters for this setting. For $8+$ sampling steps we get similar results for our alternating optimization version (Section 3.1) and the instant 2-batch version (Section 3.2) of our method. For fewer sampling steps, the alternating version performs better.

We find that our distilled models also perform very well in terms of Inception Score (Salimans et al., 2016) even though we did not optimize for this. By using classifier-free guidance the Inception Score can be improved further, as we show in Section 5.3.

Table 1: Results on ImageNet 64x64.

| Method | # param | NFE | FID↓ | IS↑ |
|---|---|---|---|---|
| VDM++ (Kingma & Gao, 2023) | 2B | 1024 | 1.43 | 64 |
| RIN (Jabri et al., 2023) | 281M | 1000 | 1.23 | 67 |
| our base model | 400M | 1024 | 1.42 | 84 |
| DDIM (Song et al., 2021a) | | 10 | 18.7 | |
| TRACT (Berthelot et al., 2023) | | 1 | 7.43 | |
| | | 2 | 4.97 | |
| | | 4 | 2.93 | |
| | | 8 | 2.41 | |
| CD (LPIPS) (Song et al., 2023) | | 1 | 6.20 | |
| | | 2 | 4.70 | |
| | | 3 | 4.32 | |
| iCT-deep (Song & Dhariwal, 2023) | | 1 | 3.25 | |
| | | 2 | 2.77 | |
| PD (Salimans & Ho, 2022) | 400M | 1 | 10.7 | |
| (reimpl. from Heek et al. (2024)) | | 2 | 4.7 | |
| | | 4 | 2.4 | |
| | | 8 | 1.7 | 63 |
| MultiStep-CD (Heek et al., 2024) | 1.2B | 1 | 3.2 | |
| | | 2 | 1.9 | |
| | | 4 | 1.6 | |
| | | 8 | 1.4 | 73 |
| CTM (Kim et al., 2024) | | 2 | 1.73 | 64 |
| DMD (Yin et al., 2023) | | 1 | 2.62 | |
| Diff-Instruct (Luo et al., 2023) | | 1 | 5.57 | |
| **Moment Matching** | 400M | | | |
| Alternating (c.f. Sect. 3.1) | | 1 | 3.0 | 89 |
| | | 2 | 3.86 | 60 |
| | | 4 | 1.50 | 75 |
| | | 8 | **1.24** | 78 |
| Instant (c.f. Sect. 3.2) | | 4 | 3.4 | **98** |
| | | 8 | 1.35 | 81 |

Table 2: Results on ImageNet 128x128

| Method | # param | NFE | FID↓ | IS↑ |
|---|---|---|---|---|
| VDM++ (Kingma & Gao, 2023) | 2B | 1024 | 1.75 | 171 |
| our base model | 400M | 1024 | 1.76 | 194 |
| PD (Salimans & Ho, 2022) | 400M | 2 | 8.0 | |
| (reimpl. from Heek et al. (2024)) | | 4 | 3.8 | |
| | | 8 | 2.5 | 162 |
| MultiStep-CD (Heek et al., 2024) | 1.2B | 1 | 7.0 | |
| | | 2 | 3.1 | |
| | | 4 | 2.3 | |
| | | 8 | 2.1 | 160 |
| **Moment Matching** | 400M | | | |
| Alternating (c.f. Sect. 3.1) | | 1 | 3.3 | 170 |
| | | 2 | 3.14 | 163 |
| | | 4 | 1.72 | 184 |
| | | 8 | **1.49** | 184 |
| Instant (c.f. Sect. 3.2) | | 4 | 3.48 | **232** |
| | | 8 | 1.54 | 183 |

**How can a distilled model improve upon its teacher?** On Imagenet our distilled diffusion model with 8 sampling steps and no classifier-free guidance outperforms its 512-step teacher with optimized guidance level, for both the $64 \times 64$ and $128 \times 128$ resolution. This result might be surprising since the many-step teacher model is often seen as the gold standard for sampling quality. However, even the teacher model has prediction error that makes it possible to improve upon it. In theory, predictions of the clean data at different diffusion times are all linked and should be mutually consistent, but since the diffusion model is implemented with an unconstrained neural network this generally will not be the case in practice. Prediction errors will thus be different across timesteps which opens up the possibility of improving the results by averaging over these predictions in the right way. Different sampling algorithms average over these predictions differently, as shown in Appendix E, which offers scope for improvement.

Similarly, prediction error will not be constant over the model inputs $\mathbf{z}_t$, and biasing generation away from areas of large error could also yield sampling improvements. Although many-step ancestral sampling typically gives good results, and is often better than deterministic samplers like DDIM, it's not necessarily optimal. In future work we hope to study the improvement of moment matching over our base sampler in further detail, and test our hypotheses about its causes.

### 5.2 Ablating conditional sampling

The distilled generator in our proposed method samples from the conditional $q(\mathbf{z}_s|\tilde{\mathbf{x}}, \mathbf{z}_t)$, whereas existing distillation methods based on score matching typically don't condition on $\mathbf{z}_t$. Instead they apply noise independently, mirroring the forward diffusion process used during training the original model. When using a 1-step sampling setup, the two approaches are equivalent since any intermediate $\mathbf{z}_s$ will be independent from the starting point $\mathbf{z}_1$ if that point has zero signal-to-noise. In the multistep setup the two approaches are meaningfully different however, and sampling from the conditional $q(\mathbf{z}_s|\tilde{\mathbf{x}}, \mathbf{z}_t)$ or the marginal $q(\mathbf{z}_s|\tilde{\mathbf{x}})$ are both valid choices. We ablate our choice of conditioning on $\mathbf{z}_t$ versus applying noise independently, and find that conditioning leads to much better sample diversity in the distilled model, as shown in Figure 3.

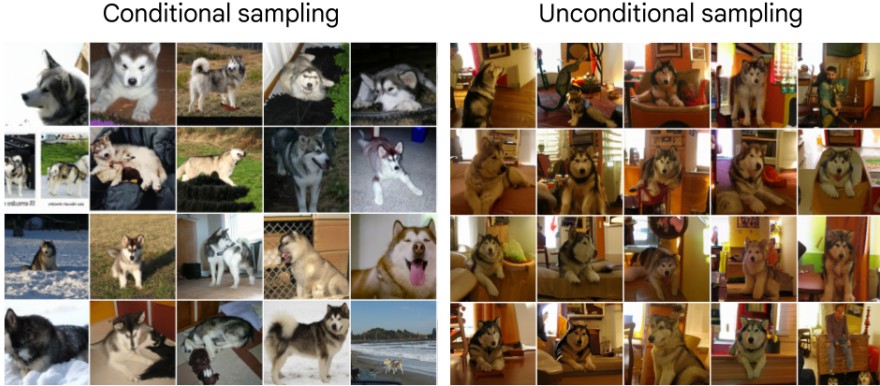

Figure 3: Multistep distillation results for a single Imagenet class obtained with two different methods of sampling from the generator during distillation: Conditional $q(\mathbf{z}_s|\tilde{\mathbf{x}}, \mathbf{z}_t)$, and unconditional $q(\mathbf{z}_s|\tilde{\mathbf{x}})$. Our choice of sampling from the conditional yields much better sample diversity.

### 5.3 Effect of classifier-free guidance

Our distillation method can be used with or without guidance. For the alternating optimization version of our method we only apply guidance in the teacher model, but not in the generator or auxiliary denoising model. For the instant 2-batch version we apply guidance and clipping to the teacher model and then calculate its gradient with a straight through approximation. Exper-

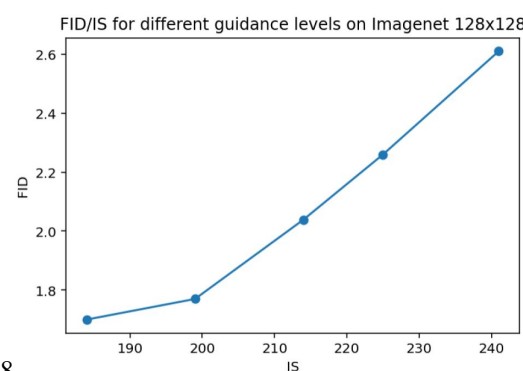

imenting with different levels of guidance, we find that increasing guidance typically increases Inception Score and CLIP Score, while reducing FID, as shown in the adjacent figure.

### 5.4 Distillation loss is informative for moment matching

A unique advantage of the instant 2-batch version of our moment matching approach is that, unlike most other distillation methods, it has a simple loss function (equation 9) that is minimized without adversarial techniques, bootstrapping, or other tricks. This means that the value of the loss is useful for monitoring the progress of the distillation algorithm. We show this for Imagenet $128 \times 128$ in the adjacent figure: The typical behavior we see is that the loss tends to go up slightly for the first few optimization steps, after which it exponentially falls to zero with increasing number of parameter updates.

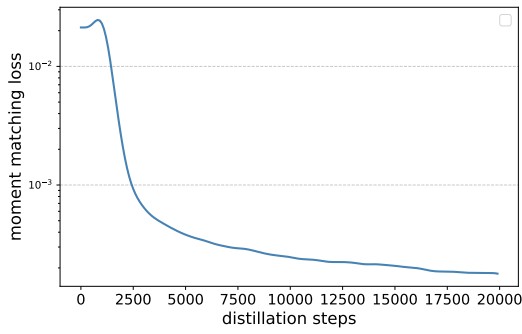

### 5.5 Text to image

To investigate our proposed method's potential to scale to large text-to-image models we train a pixel-space model (no encoder/decoder) on a licensed dataset of text-image pairs at a resolution of $512 \times 512$, using the UViT model and shifted noise schedule from Simple Diffusion (Hoogeboom et al., 2023) and using a T5 XXL text encoder following Imagen (Saharia et al., 2022). We compare the performance of our base model against an 8-step distilled model obtained with our moment matching method. In Table 3 we report zero-shot FID (Heusel et al., 2017) and CLIP Score (Radford et al., 2021) on MS-COCO (Lin et al., 2014): Also in this setting we find that our distilled model with alternating optimization exceeds the metrics for our base model. The instant 2-batch version of our algorithm performs somewhat less well at 8 sampling steps. Samples from our distilled text-to-image model are shown in Figure 1 and in Figure 7 in the appendix.

Table 3: Results on text-to-image, $512 \times 512$.

| Method | NFE | guidance | COCO FID$_{30k}$ ↓ | CLIP Score ↑ |
|---|---|---|---|---|
| our base model | 512 | 0 | 9.6 | 0.290 |
| | 512 | 0.5 | 7.9 | 0.305 |
| | 512 | 3 | 12.7 | 0.315 |
| | 512 | 5 | 13.4 | 0.316 |
| StableDiffusion v1.5[*] | 512 | low | 8.78 | |
| (Rombach et al., 2022) | 512 | high | 13.5 | **0.322** |
| DMD | 1 | low | 11.5 | |
| (Yin et al., 2023) | 1 | high | 14.9 | 0.32 |
| UFOGen | 1 | | 12.8 | 0.311 |
| (Xu et al., 2023b) | | | | |
| SwiftBrush | 1 | | 16.67 | 0.29 |
| (Nguyen & Tran, 2023) | | | | |
| InstaFlow-1.7B | 1 | | 11.8 | 0.309 |
| (Liu et al., 2023) | | | | |
| PeRFlow | 4 | | 11.3 | |
| (Yan et al., 2024) | | | | |
| **Moment Matching** | | | | |
| Alternating (Sec. 3.1) | 8 | 0 | **7.25** | 0.297 |
| | 8 | 3 | 14.15 | 0.319 |
| Instant (Sec. 3.2) | 8 | 0 | 9.5 | 0.300 |
| | 8 | 3 | 19.0 | 0.306 |

[*] Reported results for StableDiffusion v1.5 are from Yin et al. (2023).

## 6 Conclusion

We presented *Moment Matching Distillation*, a method for making diffusion models faster to sample. The method distills many-step diffusion models into few-step models by matching conditional expectations of the clean data given noisy data along the sampling trajectory. The moment matching framework provides a new perspective on related recently proposed distillation methods and allows us to extend these methods to the multi-step setting. Using multiple sampling steps, our distilled models consistently outperform their one-step versions, and often even exceed their many-step teachers, setting new state-of-the-art results on the Imagenet dataset. However, automated metrics of image quality are highly imperfect, and in future work we plan to run a full set of human evaluations on the outputs of our distilled models to complement the metrics reported here.

We presented two different versions of our algorithm: One based on alternating updates of a distilled generator and an auxiliary denoising model, and another using two minibatches to allow only updating the generator. In future work we intend to further explore the space of algorithms spanned by these choices, and gain additional insight into the costs and benefits of both approaches.

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

## A   *Instant* moment matching = matching expected teacher gradients

In section 3.2 we propose an *instantaneous* version of our moment matching loss that does not require alternating optimization of an auxiliary denoising model $g_\phi$. This alternative version of our algorithm uses the loss in equation 8, which we reproduce here for easy readibility:

$$L_{\text{instant}}(\eta) = \lim_{\lambda \to 0} \frac{1}{\lambda} L_{\phi(\lambda)}(\eta) = w(s)\tilde{\mathbf{x}}^T \text{sg}\Big\{ \frac{\partial g_\theta(\mathbf{z}_s)}{\partial \theta} \Lambda \nabla_\phi L(\phi)|_{\phi=\theta} \Big\}, \tag{15}$$

where $\frac{\partial g_\theta(\mathbf{z}_s)}{\partial \theta}$ is the Jacobian of $g_\theta$, and where $\nabla_\phi L(\phi)$ is evaluated on an independent minibatch from $\tilde{\mathbf{x}}$ and $\mathbf{z}_s$.

It turns out that this loss can be expressed as performing moment matching in teacher-parameter space rather than $\mathbf{x}$-space. Denoting the standard diffusion loss as $L_\theta(\mathbf{x}, \mathbf{z}_s) \equiv w(s)\|\mathbf{x} - g_\theta(\mathbf{z}_s)\|^2$, we can rewrite the term $\nabla_\phi L(\phi)|_{\phi=\theta} = \nabla_\theta L_\theta(\tilde{\mathbf{x}}, g_\theta(\mathbf{z}_s))$ because the first term of $L(\phi)$ can be seen as a standard diffusion loss at $\theta$ for a generated $\tilde{\mathbf{x}}$, and the second term of $L(\phi)$ is zero when $\phi = \theta$. Futher observe that $\nabla_\theta L_\theta(\mathbf{x}, \mathbf{z}_s) = 2w(s)(g_\theta(\mathbf{z}_s) - \tilde{\mathbf{x}})^T \frac{\partial g_\theta(\mathbf{z}_s)}{\partial \theta}$ which means that $\nabla_\eta \nabla_\theta L_\theta(\mathbf{x}, \mathbf{z}_s) = \nabla_\eta 2w(s)\tilde{\mathbf{x}}^T \frac{\partial g_\theta(\mathbf{z}_s)}{\partial \theta}$. Now letting $\tilde{L}_{\text{instant}}(\eta) = L_{\text{instant}}(\eta) + w(s)g_\theta(\mathbf{z}_s)^T \frac{\partial g_\theta(\mathbf{z}_s)}{\partial \theta} \nabla_\theta L_\theta(\mathbf{x}, \mathbf{z}_s)$ where the latter term is constant w.r.t. $\eta$, we can write instant moment-matching (Equation 15) as moment-matching of the teacher gradients where again stop gradients are again placed on $\mathbf{z}_s$:

$$\tilde{L}_{\text{instant}}(\eta) \equiv \frac{1}{2}\|\mathbb{E}_{\mathbf{z}_t \sim q, \tilde{\mathbf{x}}=g_\eta(\mathbf{z}_t), \mathbf{z}_s \sim q(\mathbf{z}_s|\mathbf{z}_t, \tilde{\mathbf{x}})} \nabla_\theta L_\theta(\tilde{\mathbf{x}}, \text{sg}(\mathbf{z}_s))\|_\Lambda^2 \tag{16}$$

$$= \frac{1}{2}\|\mathbb{E}_{\mathbf{z}_t \sim q, \tilde{\mathbf{x}}=g_\eta(\mathbf{z}_t), \mathbf{z}_s \sim q(\mathbf{z}_s|\mathbf{z}_t, \tilde{\mathbf{x}})} \nabla_\theta L_\theta(\tilde{\mathbf{x}}, \text{sg}(\mathbf{z}_s)) - \mathbb{E}_{\mathbf{x}, \mathbf{z}'_s \sim q} \nabla_\theta L_\theta(\mathbf{x}, \mathbf{z}'_s)\|_\Lambda^2, \tag{17}$$

where we assume that the gradient of the teacher training loss is zero when sampling from the training distribution, $\mathbb{E}_{\mathbf{x}, \mathbf{z}'_s \sim q} \nabla_\theta L_\theta(\mathbf{x}, \mathbf{z}'_s) = 0$, which is true if the teacher attained a minimum of its training loss. Our instant moment matching variant is thus indeed equivalent to matching expected teacher gradients in parameter space.

## B   Experimental details

All experiments were run on TPUv5e, using 256 chips per experiment. For ImageNet we used a global batch size of 2048, while for text-to-image we used a global batch size of 512. The base models were trained for 1M steps, requiring between 2 days (Imagenet 64) to 2 weeks (text-to-image). We use the UViT architecture from Hoogeboom et al. (2023). Configurations largely correspond to those in the appendix of Hoogeboom et al. (2023), where we used their small model variant for our Imagenet experiments.

For Imagenet we distill the trained base models for a maximum of 200,000 steps, and for text-to-image we use a maximum of 50,000 steps. We report the best FID obtained during distillation, evaluating every 5,000 steps. We fix the random seed and data used in each evaluation to minimize biasing our results. We use the Adam optimizer (Kingma & Ba, 2014) with $\beta_1 = 0, \beta_2 = 0.99, \epsilon = 1e^{-12}$. We use learning rate warmup for the first 1,000 steps and then linearly anneal the learning rate to zero over the remainder of the optimization steps. We use gradient clipping with a maximum norm of 1. We don't use an EMA, weight decay, or dropout.

## C   More model samples

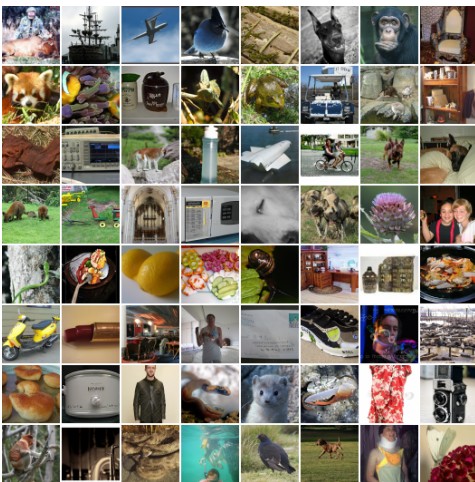

Figure 4: Random samples for random ImageNet classes at the $64 \times 64$ resolution, from our 8-step distilled model, using the alternating optimization version of our algorithm.

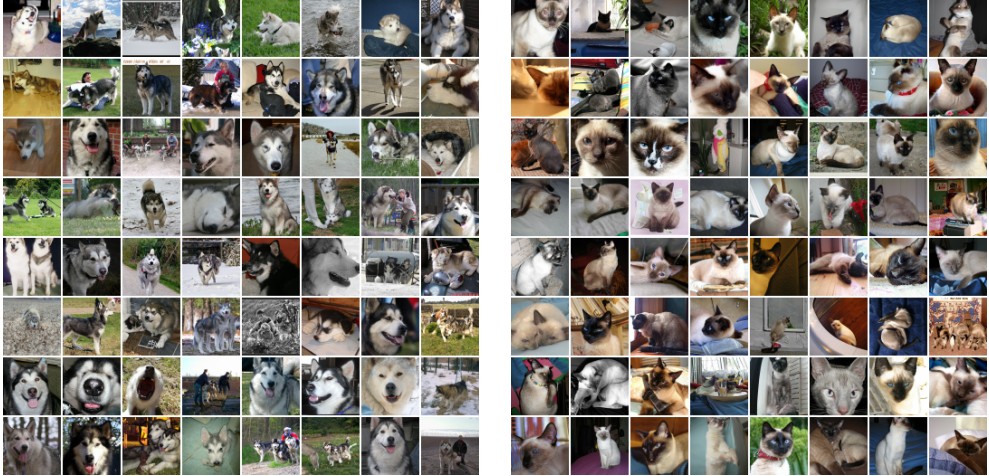

Figure 5: Random samples for a single ImageNet class at the $64 \times 64$ resolution, from our 8-step distilled model. Visualizing samples from a single class helps to assess sample diversity.

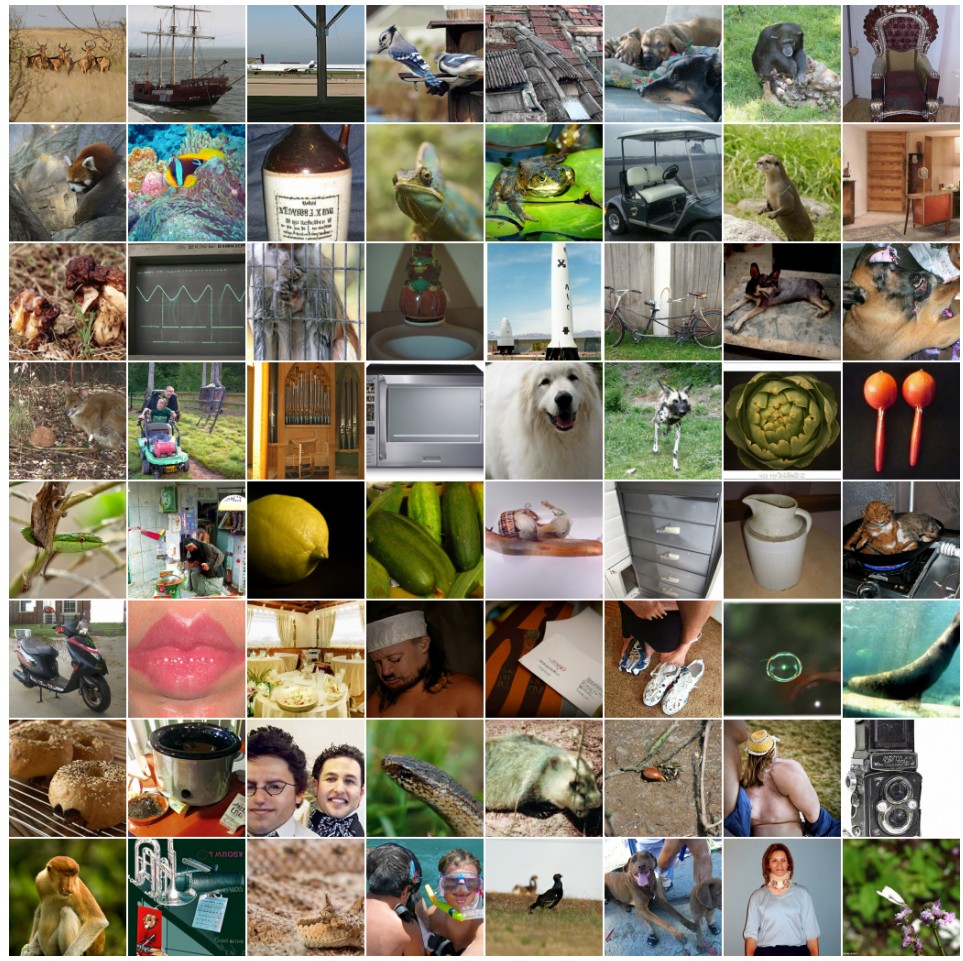

Figure 6: Random samples for random ImageNet classes at the $128 \times 128$ resolution from our 8-step distilled model, using the alternating optimization version of our algorithm.

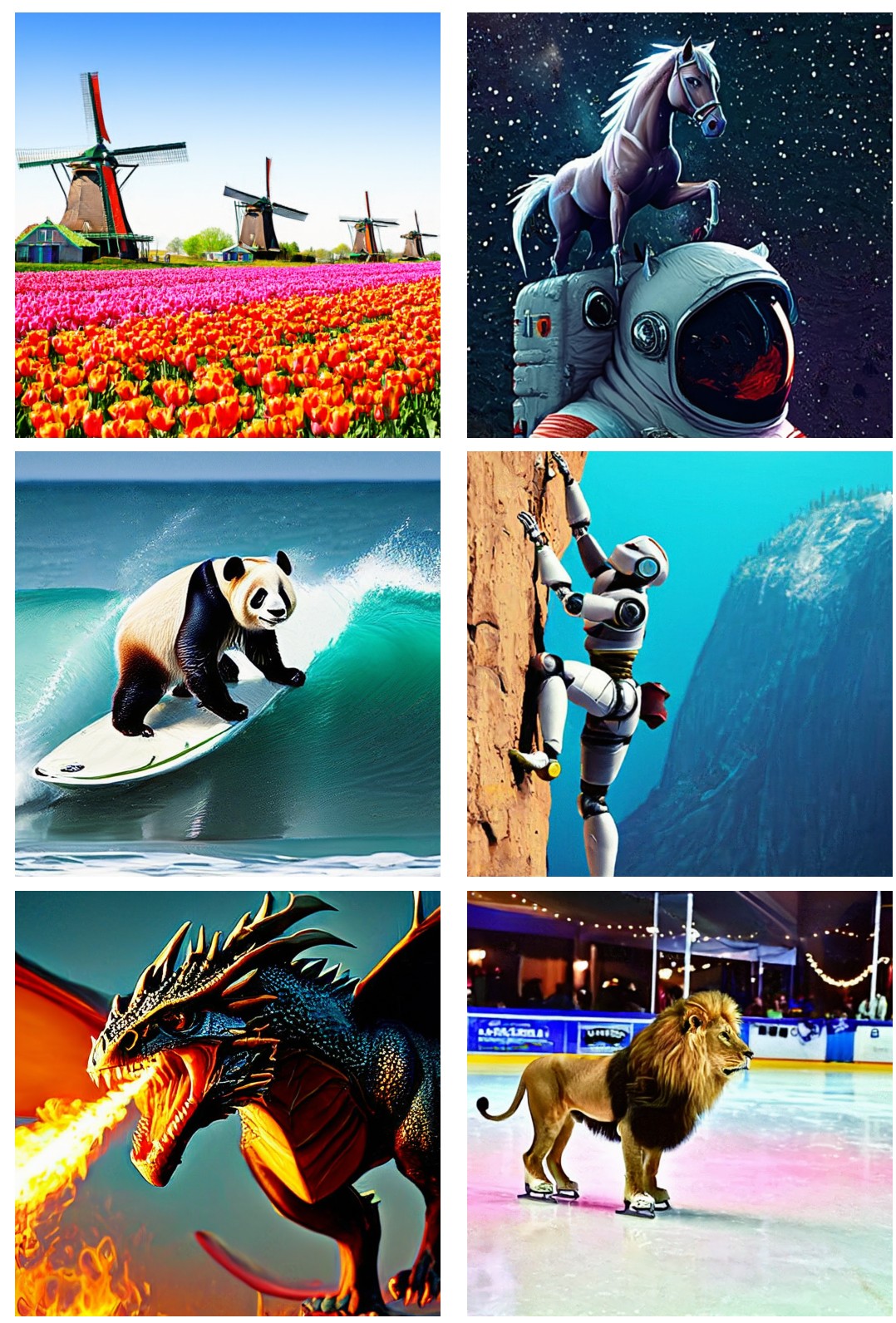

Figure 7: Selected 8-step samples from our distilled text-to-image model.

## D Relationship to score matching

When distilling a diffusion model using moment matching with alternating parameter updates (Section 3.1) the auxiliary denoising model $\hat{\mathbf{x}}_\phi$ is fit by minimizing $\mathbb{E}_{p_\eta} \tilde{w}(\mathbf{z}_s) \| \tilde{\mathbf{x}} - \hat{\mathbf{x}}_\phi(\mathbf{z}_s) \|^2$, with $\mathbf{z}_s$ and $\tilde{\mathbf{x}}$ sampled from our distilled model. In the one-step case this is equivalent to performing score matching, as $\tilde{\mathbf{x}} = g_\eta(\mathbf{z}_1)$ with $\mathbf{z}_1 \sim N(0, \boldsymbol{I})$, and $\mathbf{z}_s \sim q(\mathbf{z}_s | \tilde{\mathbf{x}})$ is sampled using forward diffusion starting from $\tilde{\mathbf{x}}$. Recall here that $q(\mathbf{z}_s | \tilde{\mathbf{x}}, \mathbf{z}_1) = q(\mathbf{z}_s | \tilde{\mathbf{x}})$ as $\mathbf{z}_1$ is pure noise, uncorrelated with $\mathbf{z}_s$. Upon convergence, we'll have that

$$(\alpha_s \hat{\mathbf{x}}_\phi(\mathbf{z}_s) - \mathbf{z}_s)/\sigma_s^2 = \mathbb{E}_{p_\eta(\tilde{\mathbf{x}}|\mathbf{z}_s)}[(\alpha_s \tilde{\mathbf{x}} - \mathbf{z}_s)/\sigma_s^2] = \nabla_{\mathbf{z}_s} \log p_\eta(\mathbf{z}_s) \, \forall \mathbf{z}_s,$$

i.e. our auxiliary model will match the score of the marginal sampling distribution of the distilled model $p_\eta(\mathbf{z}_s)$ (because the optimal solution is $\hat{\mathbf{x}}_\phi(\mathbf{z}_s) = \mathbb{E}_{\tilde{\mathbf{x}} \sim p_\eta}[\tilde{\mathbf{x}}]$). In the multi-step case this equivalence does not hold, since the sampling distribution of $\mathbf{z}_s$ depends on $\mathbf{z}_t$. Instead the proper *multistep score* is given by:

$$\nabla_{\mathbf{z}_s} \log p_\eta(\mathbf{z}_s) = \mathbb{E}_{p_\eta(\tilde{\mathbf{x}}, \mathbf{z}_t | \mathbf{z}_s)}[\nabla_{\mathbf{z}_s} \log p_\eta(\mathbf{z}_s | \tilde{\mathbf{x}}, \mathbf{z}_t)] \tag{18}$$

$$= \mathbb{E}_{p_\eta(\tilde{\mathbf{x}}, \mathbf{z}_t | \mathbf{z}_s)}[(\boldsymbol{\mu}_{t \to s}(\tilde{\mathbf{x}}, \mathbf{z}_t) - \mathbf{z}_s)/\sigma_{t \to s}^2], \tag{19}$$

$$\text{with } \sigma_{t \to s}^2 = \left( \frac{1}{\sigma_s^2} + \frac{\alpha_{t|s}^2}{\sigma_{t|s}^2} \right)^{-1} \text{ and } \boldsymbol{\mu}_{t \to s} = \sigma_{t \to s}^2 \left( \frac{\alpha_{t|s}}{\sigma_{t|s}^2} \mathbf{z}_t + \frac{\alpha_s}{\sigma_s^2} \tilde{\mathbf{x}} \right). \tag{20}$$

This expression suggests we could perform score matching by denoising towards $\boldsymbol{\mu}_{t \to s}(\tilde{\mathbf{x}}, \mathbf{z}_t)$, a linear combination of $\tilde{\mathbf{x}}$ and $\mathbf{z}_t$, rather than just towards $\tilde{\mathbf{x}}$. We tried this in early experiments, but did not get good results.

However, moment matching can still be seen to match the proper score expression (equation 19) *approximately*, if we assume that the forward processes match, meaning $p_\eta(\mathbf{z}_t | \mathbf{z}_s) \approx q(\mathbf{z}_t | \mathbf{z}_s)$. This then gives:

$$\nabla_{\mathbf{z}_s} \log p_\eta(\mathbf{z}_s) \approx \mathbb{E}_{p_\eta(\tilde{\mathbf{x}}|\mathbf{z}_s)q(\mathbf{z}_t|\mathbf{z}_s)}[(\boldsymbol{\mu}_{t \to s}(\tilde{\mathbf{x}}, \mathbf{z}_t) - \mathbf{z}_s)/\sigma_{t \to s}^2] \tag{21}$$

$$= \mathbb{E}_{p_\eta(\tilde{\mathbf{x}}|\mathbf{z}_s)q(\mathbf{z}_t|\mathbf{z}_s)} \left[ \frac{\alpha_{t|s}}{\sigma_{t|s}^2} \mathbf{z}_t + \frac{\alpha_s}{\sigma_s^2} \tilde{\mathbf{x}} - \left( \frac{1}{\sigma_s^2} + \frac{\alpha_{t|s}^2}{\sigma_{t|s}^2} \right) \mathbf{z}_s \right] \tag{22}$$

$$= \mathbb{E}_{p_\eta(\tilde{\mathbf{x}}|\mathbf{z}_s)} \left[ \frac{\alpha_{t|s}^2}{\sigma_{t|s}^2} \mathbf{z}_s + \frac{\alpha_s}{\sigma_s^2} \tilde{\mathbf{x}} - \left( \frac{1}{\sigma_s^2} + \frac{\alpha_{t|s}^2}{\sigma_{t|s}^2} \right) \mathbf{z}_s \right] \tag{23}$$

$$= \mathbb{E}_{p_\eta(\tilde{\mathbf{x}}|\mathbf{z}_s)} \left[ \frac{\alpha_s}{\sigma_s^2} \tilde{\mathbf{x}} - \frac{1}{\sigma_s^2} \mathbf{z}_s \right] \tag{24}$$

$$= (\alpha_s \mathbb{E}_{p_\eta}[\tilde{\mathbf{x}} | \mathbf{z}_s] - \mathbf{z}_s)/\sigma_s^2. \tag{25}$$

When this is used to fit the auxiliary score $s_\phi(\mathbf{z}_s) = (\alpha_s \hat{\mathbf{x}}_\phi(\mathbf{z}_s) - \mathbf{z}_s)/\sigma_s^2$, it is equivalent to fitting $\hat{\mathbf{x}}_\phi(\mathbf{z}_s)$ against just $\tilde{\mathbf{x}}$, so under this approximation moment matching and score matching once again become equivalent.

If our distilled model has $p_\eta(\tilde{\mathbf{x}} | \mathbf{z}_t) = q(\mathbf{x} | \mathbf{z}_t)$, and if $\mathbf{z}_t \sim q(\mathbf{z}_t)$ (which is true during training), the approximation $p_\eta(\mathbf{z}_t | \mathbf{z}_s) \approx q(\mathbf{z}_t | \mathbf{z}_s)$ would become exact. Both multistep moment matching and multistep score matching thus have a fixed point that corresponds to the correct target distribution $q$. We currently do not have any results on guaranteeing when this fixed point is indeed attained for both methods, and exploring this further would make for useful future research. Note that in general $p_\eta(\mathbf{z}_t | \mathbf{z}_s) \neq q(\mathbf{z}_t | \mathbf{z}_s)$ until convergence, so during optimization moment matching and score matching indeed optimize different objectives.

## E How different sampling algorithms average over model predictions

In Section 5 we note that our distilled models often improve over their teacher, and we speculate that this may be due to the way the student model averages over teacher predictions. In theory, predictions of the clean data at different diffusion times should be mutually consistent, but since the diffusion model is implemented with an unconstrained neural network this generally will not be the case in practice. Different sampling algorithms average over these predictions differently, which offers scope for improvement if a more optimal weighting can be found.

Sampling from a diffusion models consists of repeatedly predicting the clean data given an intermediate noisy data point. Ignoring the dependence between these predictions (earlier predictions determine the input of later predictions), we can say that the final sample is then a simple weighted average of these predictions. In Figure E we show how the standard ancestral (DDPM) sampler (Ho et al., 2020) and the deterministic DDIM sampler (Song et al., 2021a) perform this weighting, where our neural network uses the *v-parameterization* of Salimans & Ho (2022).

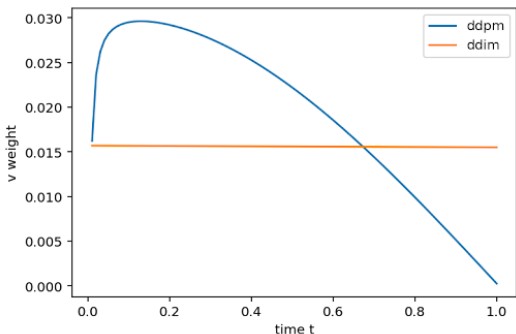

Figure 8: Implied weighting of v-predictions made during sampling in determining the final sample, for the DDPM and DDIM sampling algorithms.

