# OpenReview forum: "Multistep Distillation of Diffusion Models via Moment Matching"
_NeurIPS.cc/2024/Conference — NeurIPS 2024 poster_

### Official Review · Reviewer_V8xX · 2024-06-25

**Soundness:** 3
**Presentation:** 3
**Contribution:** 3
**Rating:** 6
**Confidence:** 3

**Summary:**

This paper presents a new method for making diffusion models faster to sample. The method distills many-step diffusion models into few-step models and extends recently proposed one-step methods to the multistep case by moment matching. By using up to 8 sampling steps, the obtained distilled models outperform not only their one-step versions but also their original many-step teacher models, obtaining new SOTA results on the Imagenet dataset.

**Strengths:**

1. This paper presents a new method for making diffusion models faster to sample. The method distills many-step diffusion models into few-step models and extends recently proposed one-step methods to the multistep case by moment matching.
2. The paper is well writing. The proposed moment matching proposed in this paper is a technique worthy of reference and further in-depth study by researchers.
3. Experimental results show the effectiveness of the proposed method and the results of experiments achieve SOTA.

**Weaknesses:**

1. Moment Matching Distillation Part is difficult and too mathematical to understand. The authors could improve the writing to make it easier for the reader to understand.
2. In line 446, "because the ﬁrst term of $L( \phi)$ can be seen as a standard diffusion loss at $\theta$." I don't understand, the author needs to give more interpretations.
2. The experiment of Class-conditional generation is only on ImageNet. The authors could provide more experiment results on other datasets, such as celeba, lsun and so on in later versions.
3. The experiments lack ablating sampling steps. The authors could provide more experiment results adapting other sampling steps in later versions.

**Questions:**

1. the analysis of Ablating conditional sampling is confusing. It’s not unusual that conditioning leads to much better sample diversity in the distilled model because you use extra label information. Could the authors explain more about the ablating conditional sampling?
2. Why you select 8 sampling steps? How more or less eight steps will affect experimental results, run speeds, etc?

**Limitations:**

A limitation of this study is that the theoretical part is difficult and too mathematical to understand.

---

> ### Author Rebuttal · Authors · 2024-08-03
>
> Thanks for your review. Please find our response to your comments below:
>
> > Moment Matching Distillation Part is difficult and too mathematical to understand.
>
> Appendix A of the paper contains the mathematical details we felt we could reasonably omit from the main text. In our updated version of the paper we’ll have another mathematical appendix with some of the more tangential mathematical analysis. The remaining math (e.g. the definition of the sampling distributions) we think is essential to describing the core contributions of the paper. Apart from the method section we usually describe things in words rather than math. If you have concrete suggestions for making things easier to understand we’d be very happy to consider them.
>
> > In line 446, "because the ﬁrst term of $L(\phi)$ can be seen as a standard diffusion loss at $\theta$." I don't understand, the author needs to give more interpretations.
>
> This refers to the loss $L(\phi)$ as defined in Algorithm 2. The first term is a standard diffusion loss, which is evaluated at $\phi = \theta$, i.e. using the parameters of the pretrained teacher model. We’ll make this easier to read by repeating the definition for $L(\phi)$ and by referring to Algorithm 2.
>
> > The experiment of Class-conditional generation is only on ImageNet. The authors could provide more experiment results on other datasets, such as celeba, lsun and so on in later versions.
>
> We could easily add one of these additional datasets to the appendix of the updated paper, but in our experience they are not very informative as these datasets are of much lower diversity than Imagenet and are more prone to overfitting. Please note that the paper does include experiments on a large high-resolution text-to-image dataset (section 5.5) which we think is more informative.
>
> > The experiments lack ablating sampling steps. The authors could provide more experiment results adapting other sampling steps in later versions.
>
> The paper currently presents results using 1, 2, 4, and 8 sampling steps. Are you suggesting we add results for 16 steps and higher? Qualitatively, results for between 16 and 1024 sampling steps lie somewhere in between the presented results for our distilled model at 8 sampling steps and the undistilled baseline model at 1024 steps. We could add some discussion on this if useful.
>
> > The analysis of Ablating conditional sampling is confusing. It’s not unusual that conditioning leads to much better sample diversity in the distilled model because you use extra label information. Could the authors explain more about the ablating conditional sampling?
>
> There seems to be a misunderstanding here: The experiment in Section 5.2 ablates conditioning on intermediate steps z_t, which is something that distinguishes our approach from other methods in the literature. We do not ablate conditioning on label information.
>
> > Why you select 8 sampling steps? How more or less eight steps will affect experimental results, run speeds, etc?
>
> We obtain the best results for our method using 8 sampling steps, but the results tables also report using 1, 2, and 4 steps. Results for 4 steps are still very competitive. Speed / running time scales linearly with the number of steps.

---

> > ### Comment · Reviewer_V8xX · 2024-08-11
> >
> > I thank the authors for their response. Consequently, I have raised my score.

---

### Official Review · Reviewer_cU8D · 2024-07-08

**Soundness:** 2
**Presentation:** 3
**Contribution:** 3
**Rating:** 6
**Confidence:** 5

**Summary:**

This paper proposes an approach to distill a diffusion model into a multi-step generator. Building on previous works that use distribution matching to train a few-step student generator, the paper introduces a novel method of matching the conditional expectation of clean data given noisy data along the sampling trajectory. By focusing on the matching of the first-order moment, the authors derive a gradient for training the generator, which can be approximated using two diffusion models (as seen in Diff-Instruct, DMD, and SwiftBrush) or estimated using batch statistics. The primary contribution is conditioning on noisy data during expectation matching, distinguishing this approach from previous score distillation methods that match unconditioned expectations. Extensive results demonstrate the effectiveness of this method.

**Strengths:**

S1. This paper provides a new perspective on recent distribution matching-based diffusion distillation approaches such as Diff-Instruct, DMD, and SwiftBrush. The moment matching formulation is novel and theoretically sound, leading to practical improvements including conditioning on noisy input to the generator, as well as a different way of training the model that doesn't rely on an auxiliary diffusion model

S2. The experimental validation is comprehensive, with the 8-step model achieving exceptionally strong image generation results.

**Weaknesses:**

Minor Typo: There is a typo on line 72: "as described is described."

Lack of Detailed Ablation Studies: There are several instances where the authors claim one formulation is better based on their trials but do not provide exact ablation studies. Including these studies would help readers understand the effectiveness of alternative formulations. Specific examples include:

- Exclusion of the z_s dependence on the parameters when calculating gradients of the moment matching loss (line 90).
- Training the auxiliary diffusion model using a mix of teacher predictions and generated samples (line 110).

Comparison Between Optimization Setups: Further comparison between the alternating and instant optimization setups would be beneficial. Currently, alternating optimization appears better, but instant optimization feels more principled. The authors should discuss this discrepancy in more detail and highlight potential challenges to improve performance.

Enhancement of Conditional Sampling Ablation: The ablation study on the importance of conditional sampling could be clearer. The paper shows that without conditioning on noisy input, the generator produces less diverse images with drifted tones. However, other methods like Diff-Instruct, DMD, and SwiftBrush, which don't condition on noisy input, work without these diversity/tone issues. This seems contradictory to existing literature. Does this issue arise as the number of sampling steps increases? What is the failure point for other unconditional methods? These experiments and discussions are crucial, as this is the paper's most significant contribution and claim.

On L181, the authors mention that "One could imagine fitting x through score matching against the conditional distribution q(z_s | z_t, x_~) but this did not work well when we tried it." Could the authors elaborate on "score matching against the conditional distribution"?

**Questions:**

Please find a set of suggestions in the weakness section above.

**Limitations:**

Yes.

---

> ### Author Rebuttal · Authors · 2024-08-03
>
> Thanks for your review and kind words. Please find our response to your comments below:
>
>
> * Requests for additional ablations:
>
> > Exclusion of the z_s dependence on the parameters when calculating gradients of the moment matching loss (line 90).
>
> We’d be happy to include this ablation in the camera ready update of the paper.
>
> > Training the auxiliary diffusion model using a mix of teacher predictions and generated samples (line 110).
>
> We are experimenting with this for video and other modalities, but consider this to be out of scope for the current paper.
>
> * Additional comparison between alternating and instant optimization setups.
> > Further comparison between the alternating and instant optimization setups would be beneficial. Currently, alternating optimization appears better, but instant optimization feels more principled. The authors should discuss this discrepancy in more detail and highlight potential challenges to improve performance.
>
> When taking very few sampling steps, the main discrepancy is that the instant approach drops modes more quickly due to the conditioning of the optimization problem. For 8+ steps both approaches are very similar. We can add some additional discussion on this to the paper.
>
> > The ablation study on the importance of conditional sampling could be clearer. The paper shows that without conditioning on noisy input, the generator produces less diverse images with drifted tones. However, other methods like Diff-Instruct, DMD, and SwiftBrush, which don't condition on noisy input, work without these diversity/tone issues. This seems contradictory to existing literature.
>
> In our experience, methods from the literature (e.g. Diff-Instruct, DMD) are actually extremely sensitive to hyperparameters, weighting functions, parameter initialization etc. They work for a much smaller range of hyperparameters than our multistep distillation method, and even runs that produce the FIDs reported in the literature often need to be stopped early because they eventually still end in mode collapse. A full investigation of the stability of these other methods is out of scope for this paper, but we could add some more discussion on this if useful.
>
> > On L181, the authors mention that "One could imagine fitting $\hat{x}_{\phi}$ through score matching against the conditional distribution $q(z_s | z_t, \tilde{x})$ but this did not work well when we tried it." Could the authors elaborate on "score matching against the conditional distribution"?
>
> In standard score matching, like used in DDPM and Diff-Instuct/DMD, we minimize the average L2 distance between an estimate score $s_{\phi}(z_s)$ and the score of the forward process $\nabla_{z_s} \log q(z_s | \tilde{x})$. One way of extending this to the conditional multistep setting would be to make both expressions conditional on $z_t$, i.e. minimize the average L2 between $s_{\phi}(z_s; z_t)$ and $\nabla_{z_s} \log q(z_s | z_t, \tilde{x})$. In early experiments we did not find this to work well for distillation, so we chose to go with the current moment matching objective instead, which is fundamentally different. In the updated version of the paper we have an appendix with a more detailed discussion of the relationship to score matching. (This is already finished, but NeurIPS does not allow us to add new math/text content at this stage of the reviewing process)

---

> > ### Comment · Reviewer_cU8D · 2024-08-07
> >
> > I thank the authors for their response. I will keep my original rating.

---

### Official Review · Reviewer_Xksr · 2024-07-10

**Soundness:** 3
**Presentation:** 2
**Contribution:** 4
**Rating:** 6
**Confidence:** 3

**Summary:**

The authors proposed a fast sampling method by distilling a diffusion model to model $q(x|z_t)$. This is achieved by matching moments, with two novel approaches proposed to implement it in practice.

**Strengths:**

- The idea of distilling to model the conditional distribution $q(x|z_t)$ by using matching the moments is interesting and novel.

- The equivalence between the proposed instantaneous version of the moment matching loss and matching expected teacher gradients is interesting and insightful.

**Weaknesses:**

1. Some of the notations should be clarified or made coherent. For instance, do both $g_{\theta}(z_t, t)$ and $g_{\theta}(z_t)$ indicate the parametrized denoiser? What's the definition of $q(z_t)$ in Eq. (6)? In Eq. (4) and Eq. (6), what depends on $\eta$ (please be consistent about dropping this parameter)?

2. Algorithm 2 is somehow heuristics. It may be hard to see if $\mathbb{E}_{g}[\tilde{x}|z_s]$ can be approximated well with the auxiliary network model.

**Questions:**

1. In tables 1 and 2, by increasing NFE to the level of the base model, what the FIDs and IS will be?

2. I understand that the sampling in the base diffusion model may not be optimal. However, which mechanisms in the proposed methods can achieve better FID than the base model with distillation?

3. How is the generation quality and FID by using instantaneous version of the moment matching (algorithm 3) on ImageNet $128\times128$?

4. Can the proposed method be compatible with Consistency Model for further timestep distillation?

5. Also, can the method support training from scratch?

6. In Algorithm 3, storing and computing both teacher gradients and $\eta$-gradients appears costly, particularly as the model's scale (in terms of trainable parameters) increases. I'm curious about the scalability of this distillation method to large latent diffusion models.

**Limitations:**

The authors provided a discussion on potential limitations on evaluations and 1 or 2 steps generation.

---

> ### Author Rebuttal · Authors · 2024-08-03
>
> Thanks for your review. Please find our response to your comments and questions here:
>
> > Notation
>
> Thanks! We’ll clarify the notation issues you identified in the paper.
> $g_{\theta}(z_t, t)$ and $g_{\theta}(z_t)$ indeed refer to the same denoising model. Dropping the dependence on $t$ is customary, but we’ll make this explicit in the text.
> q() refers to the forward diffusion process, i.e. $q(z_t) = \int q(z_t | x) q(x) dx$ as defined in equation 3. We’ll add a comment on this after equation 6.
> Dependency on $\eta$ in equations 4 and 6 is through $g$. We’ll make this explicit.
>
>
> > It may be hard to see if $\mathbb{E}_{g}[\tilde{x}|z_s]$ can be approximated well with the auxiliary network model.
>
> This is indeed an approximation. However since this auxiliary denoising task is almost identical to the original denoising problem, just using different samples, we do not expect this to be a practical limitation.
>
> > In tables 1 and 2, by increasing NFE to the level of the base model, what the FIDs and IS will be?
>
> As the NFE is increased to the level of the base model, the FID and IS converge to those of the base model. For sufficiently large NFE, the distillation loss becomes zero at initialization (starting from the parameters of the base model) and hence distillation does not change the model.
>
> > Which mechanisms in the proposed methods can achieve better FID than the base model with distillation?
>
> We hypothesize the improvement with respect to ancestral sampling on the base model to derive from having a different implied weighting on the denoising predictions made at different timesteps during sampling. The final sample $x$ produced by a diffusion model sampler can be written as a linear combination of predictions made along the sampling path. For different types of samplers the weighting of these predictions look different: e.g. deterministic DDIM sampling puts more weight on the first few predictions compared to stochastic ancestral sampling (also see the discussion in the Karras et al. EDM paper on this topic). Our distillation method trains a student model that similarly tries to match an implicit weighted average of teacher model predictions, with a weighting that apparently works better than the standard samplers. Of course this is all dependent on the chosen hyperparameters and sampling schedule, and for different choices our method could potentially perform less well than standard samplers applied to the base model. Our explanation here is speculative, and we do not know of a simple experiment that would clearly confirm this, but we’d be happy to include some of this discussion in the paper.
>
> > How is the generation quality and FID by using instantaneous version of the moment matching (algorithm 3) on ImageNet 128x128?
>
> These results are shown at the bottom of Table 2 in the paper
>
> > Can the proposed method be compatible with Consistency Model for further timestep distillation?
>
> One possibility would be to first distill a diffusion model using consistency distillation, and then further improve it by finetuning with our moment matching objective. Another reviewer may have asked for an experiment like this (if we understand them correctly), so we’re likely to include this in the updated paper. Would this be of interest?
>
> > Can the method support training from scratch?
>
> The proposed distillation methods (as well as other related methods from the literature) are sensitive to the initialization of the student, and do not give good results when training from scratch. We hope that future work can improve this class of methods further to enable training from scratch.
>
> > In Algorithm 3, storing and computing both teacher gradients and $\eta$ gradients appears costly, particularly as the model's scale (in terms of trainable parameters) increases. I'm curious about the scalability of this distillation method to large latent diffusion models.
>
> Regarding storage (memory): Since gradients and parameters can be fully sharded over devices (see e.g. the ZeRO optimizer work) this is not a bottleneck in practice. In our experiments we went up to 5B parameters, using small-memory accelerators (TPU v5e), and we could have scaled even further. Regarding compute: A full iteration of algorithm 3 costs approximately the equivalent of 8 forward passes through the diffusion model (2x forward on student, fwd + bwd on the teacher for batch 1, fwd on the teacher for batch 2, backward on student for batch 2, where full backward passes are generally about twice as expensive as forward passes). This scaling does not change with the number of parameters. In absolute terms this makes one iteration of our method about equally expensive as the approaches in DMD and DiffInstruct, and about 60% more expensive than one iteration of Progressive Distillation. We could add some discussion on this to the paper if useful.

---

> ### Comment · Reviewer_Xksr · 2024-08-12
> **Thanks for clarification.**
>
> I appreciate the reviewers' clarification and support the acceptance of this paper. In the revised edition, it would be beneficial to discuss the "scalability of this distillation method to large latent diffusion models" and to further investigate "why the student outperforms the teacher".

---

> > ### Author Response · Authors · 2024-08-13
> > **Thanks. We'll include both points in the revised paper.**
> >
> > Thanks. We'll include both points in the revised paper.

---

### Official Review · Reviewer_14eo · 2024-07-12

**Soundness:** 3
**Presentation:** 3
**Contribution:** 2
**Rating:** 6
**Confidence:** 3

**Summary:**

This paper proposes a diffusion distillation algorithm based on moment matching. The starting point of the paper is to achieve distribution consistency by ensuring that the denoising process $\widetilde{x} = g_\eta(z_t,t)$ with fewer steps conforms to the true distribution, incorporating moment matching. To estimate the first moment of the student models' $x_0$ predictions, the paper suggests using a small surrogate denoising network to learn moment information. Simultaneously, the moment information from the surrogate denoising network is used to optimize the student models. The distillation model proposed in this paper demonstrates good performance with 8-step sampling

**Strengths:**

1. Research based on moment matching requires moment information. This paper proposes a surrogate denoising network for the first moment, based on the principle that one-step sampling in diffusion is unbiased for the first moment. The distillation model needed is trained using this moment network. The approach of using this surrogate is both reasonable and solid

2. The experimental section of the paper demonstrates that, compared to different distillation models, the MM-based distillation model achieves the best results with 8-step sampling

**Weaknesses:**

1. Theoretically, according to the current setting of the paper, the discretization error in the distillation model sampling originates from the gap between the one-step prediction of the distillation model and the first moment of the true teacher models' denoising. It is worth exploring the values of L(ϕ)  at different timesteps after training.

2. The experiments revealed that the performance of the distillation model worsens as the number of steps increases after training (FID of 3.0 for one step and 3.86 for two steps). Could you provide a theoretical explanation for this phenomenon?

3. For the distillation model, the number of steps in the solver of pretrained diffusion models significantly affects the results. This paper lacks an analysis of the solvers' step counts, such as the impact on results when reducing from DDIM 50 steps to DDIM 25 steps.

4. Theoretically, the method proposed in this paper can be applied to any distillation model. If the distillation model uses a pre-trained distillation model for initialization, the accuracy of the x0x0​ predictions will be higher in the early stages of training. Consequently, the optimization effect of the moment estimation surrogate network will be better. Further fine-tuning on this basis could additionally demonstrate the extent to which moment matching improves the distillation model. Could this experiment be supplemented？

**Questions:**

Please see Weakness.

**Limitations:**

yes

---

> ### Author Rebuttal · Authors · 2024-08-03
>
> Thanks for your review. Please find our response to your remarks below:
>
>
> > 1. [...] It is worth exploring the values of L(ϕ) at different timesteps after training.
>
> The instantaneous version of L(ϕ) is indeed informative, and we investigate its value over different training steps in section 5.4 (averaged over all sampling timesteps). Can we interpret your suggestion as adding a figure like this for each of the 4-8 sampling timesteps individually? Alternatively, we could make an interpretable version of L(ϕ) in our alternating algorithm by freezing the student model after distillation and then optimizing the auxiliary diffusion model (ϕ parameters) until convergence, which we could then again report for each individual sampling timestep. We’d be happy to include either of these with a discussion in the appendix.
>
>
> > 2. The experiments revealed that the performance of the distillation model worsens as the number of steps increases after training (FID of 3.0 for one step and 3.86 for two steps). Could you provide a theoretical explanation for this phenomenon?
>
> In general we see improved performance when increasing the number of sampling steps. The only exception to this are our results on Imagenet 64x64 when taking 1-2 steps, which yield the FIDs mentioned in your comment. The explanation here is that we find that this class of methods is sensitive to hyperparameters and random seeds when taking very few sampling steps. We did not separately optimize hyperparameters for this case. The relationship between sampling steps and performance is discussed already, but we will add some discussion on the specific Imagenet results that you pointed out.
>
> > 3. For the distillation model, the number of steps in the solver of pretrained diffusion models significantly affects the results. This paper lacks an analysis of the solvers' step counts, such as the impact on results when reducing from DDIM 50 steps to DDIM 25 steps.
>
> Here there might be a slight misunderstanding: Unlike some other distillation methods, we never run a full multi-step solver (e.g. DDIM with 50 or 25 steps) on our pretrained diffusion model during distillation. Instead, we sample from the student model and use a single-step evaluation of the pretrained model to provide a training signal to the student. Hence, the distilled model has no dependence on a choice of solver for the base model. Of course the reported baseline (undistilled) numbers in our results tables do depend on the choice of solver: Here we use a tuned 1024 step stochastic sampler to provide the strongest possible baseline for comparison. If useful we could add some results here using a faster sampler (e.g. DDIM with 25 steps). Please let us know if this is what you have in mind.
>
> > 4. Theoretically, the method proposed in this paper can be applied to any distillation model. If the distillation model uses a pre-trained distillation model for initialization, the accuracy of the x0x0​ predictions will be higher in the early stages of training. Consequently, the optimization effect of the moment estimation surrogate network will be better. Further fine-tuning on this basis could additionally demonstrate the extent to which moment matching improves the distillation model. Could this experiment be supplemented?
>
> Please clarify your suggestion for the additional experiment and we’d be happy to add it if feasible. Both the distilled (student) model and the auxiliary denoising model are currently initialized from the pretrained diffusion model. Is your suggestion to run moment matching distillation warm-started from a model distilled using another algorithm such as progressive distillation or consistency distillation? If so, that sounds like an interesting experiment we’d be happy to run.

---

> > ### Comment · Reviewer_14eo · 2024-08-11
> >
> > *"Is your suggestion to run moment matching distillation warm-started from a model distilled using another algorithm such as progressive distillation or consistency distillation?"*
> >
> > Yes, that’s precisely my idea. This isn’t a mandatory experiment, but it could further demonstrate the advantages of Moment Matching. Thank you for your response and I will raise my score to 6.

---

> > > ### Author Response · Authors · 2024-08-13
> > > **we'll add this experiment**
> > >
> > > We agree that this experiment (moment matching warm-started from another distilled model) would be interesting and have started work on it. Thanks!

---

### Official Review · Reviewer_5zBL · 2024-07-12

**Soundness:** 3
**Presentation:** 3
**Contribution:** 3
**Rating:** 7
**Confidence:** 3

**Summary:**

The paper presents a novel method for distilling diffusion models to require fewer function evaluations.  The method is based on moment-matching, and two practical algorithms are presented and evaluated.  Empirical results are state-of-the-art in few-step regimes.

**Strengths:**

The paper presents a well-motivated novel method for distillation fo diffusion models.  This method achieves impressive results on standard benchmarks, and represents a solid contribution to the state of diffusion models for image generation.  The presentation is clear and complete, and the method is well situated in the literature both through empirical comparison and discussion.

**Weaknesses:**

This is a strong paper overall, without many weaknesses I noticed.  I would have liked to see more discussion of why the 1- and 2- step regimes do not perform as well as other approaches.  Is this a characteristic of the moment-matching approach?

**Questions:**

1.  In Eqn. (6), how is $\mathbb{E}_q[x\mid z_s]$ approximated?  Is it through samples from the dataset?  If so, are there additional assumptions that need to be made?  Naively, I'd expect that, especially as $s \to 1$, the true data distribution would not necessarily be unimodal around the empirical samples in the training set.  Am I overthinking this?
1.  Why does the instantaneous version of the algorithm have no reported results for 1 and 2 steps?

**Limitations:**

Although there is no limitations section in the paper, the discussion of the limitations is well addressed throughout the paper.

---

> ### Author Rebuttal · Authors · 2024-08-03
>
> Thanks for your kind words.
> Please find our answers to your questions below:
>
> > I would have liked to see more discussion of why the 1- and 2- step regimes do not perform as well as other approaches. Is this a characteristic of the moment-matching approach?
>
> As the number of sampling steps is decreased, this class of methods becomes increasingly sensitive to its hyperparameters and parameter initialization, and the variance of results over different runs increases. The 1-step case can be made to work well if everything is tuned carefully, as shown by the DMD and DiffInstruct papers which can be interpreted as special cases of our method in this setting. Using their hyperparameters is not optimal for the multistep case however, so we chose not to focus on this. We will add some discussion on this in the revision of our paper.
>
>
> > In Eqn. (6), how is $\mathbb{E}_q[x|z_s]$ is approximated? [...] are there any additional assumptions that need to be made?
>
> This expectation is approximated with the pre-trained diffusion model, which in turn is trained by denoising samples from the dataset. No additional assumptions are required, though of course our distilled model will be affected by any errors in the teacher diffusion model that is used to approximate $\mathbb{E}_q[x|z_s]$, as distillation methods typically are.
>
> > Why does the instantaneous version of the algorithm have no reported results for 1 and 2 steps?
>
> Using the standard hyperparameters, the instantaneous algorithm does not provide competitive results when using fewer than 4 sampling steps, and can even diverge. We will add some discussion on this in the captions of the relevant results tables.

---

> > ### Comment · Reviewer_5zBL · 2024-08-08
> >
> > I'd like to thank the authors for responding to the review.  After reading other reviews and the author's responses, I continue to recommend this paper for acceptance.

---

### Decision · Program_Chairs · 2024-09-25

**Decision:**

Accept (poster)

**Comment:**

This paper proposes a new distillation method for accelerating diffusion model inference. The key contribution is the notion of distribution matching between the teacher and student models, via an approximation of moment matching. Reviewers are all positive about the idea and empirical results, reflected by unanimous votings of acceptance. The AC believes that this is a great paper and recommends acceptance.